# Anti-Inflammatory Properties of Eugenol in Lipopolysaccharide-Induced Macrophages and Its Role in Preventing β-Cell Dedifferentiation and Loss Induced by High Glucose-High Lipid Conditions

**DOI:** 10.3390/molecules28227619

**Published:** 2023-11-16

**Authors:** Esmaeel Ghasemi Gojani, Bo Wang, Dong-Ping Li, Olga Kovalchuk, Igor Kovalchuk

**Affiliations:** Department of Biological Sciences, University of Lethbridge, Lethbridge, AB T1K 3M4, Canada; esmaeel.ghasemigojan@uleth.ca (E.G.G.); bo.wang5@uleth.ca (B.W.); dongping.li@uleth.ca (D.-P.L.); olga.kovalchuk@uleth.ca (O.K.)

**Keywords:** inflammation, T2DM, macrophage, β-cell, eugenol, cell viability, high glucose, high lipid

## Abstract

Inflammation is a natural immune response to injury, infection, or tissue damage. It plays a crucial role in maintaining overall health and promoting healing. However, when inflammation becomes chronic and uncontrolled, it can contribute to the development of various inflammatory conditions, including type 2 diabetes. In type 2 diabetes, pancreatic β-cells have to overwork and the continuous impact of a high glucose, high lipid (HG-HL) diet contributes to their loss and dedifferentiation. This study aimed to investigate the anti-inflammatory effects of eugenol and its impact on the loss and dedifferentiation of β-cells. THP-1 macrophages were pretreated with eugenol for one hour and then exposed to lipopolysaccharide (LPS) for three hours to induce inflammation. Additionally, the second phase of NLRP3 inflammasome activation was induced by incubating the LPS-stimulated cells with adenosine triphosphate (ATP) for 30 min. The results showed that eugenol reduced the expression of proinflammatory genes, such as *IL-1β*, *IL-6* and *cyclooxygenase-2 (COX-2)*, potentially by inhibiting the activation of transcription factors NF-κB and TYK2. Eugenol also demonstrated inhibitory effects on the levels of NLRP3 mRNA and protein and Pannexin-1 (PANX-1) activation, eventually impacting the assembly of the NLRP3 inflammasome and the production of mature IL-1β. Additionally, eugenol reduced the elevated levels of adenosine deaminase acting on RNA 1 (ADAR1) transcript, suggesting its role in post-transcriptional mechanisms that regulate inflammatory responses. Furthermore, eugenol effectively decreased the loss of β-cells in response to HG-HL, likely by mitigating apoptosis. It also showed promise in suppressing HG-HL-induced β-cell dedifferentiation by restoring β-cell-specific biomarkers. Further research on eugenol and its mechanisms of action could lead to the development of therapeutic interventions for inflammatory disorders and the preservation of β-cell function in the context of type 2 diabetes.

## 1. Introduction

Eugenol (4-allyl-2-methoxyphenol) is a naturally occurring compound found in clove, basil, cinnamon, and nutmeg. It exhibits various pharmacological activities, including analgesic, antibacterial, antifungal, antiplasmodial, antiviral, anti-obesity, and antidiabetic properties [1].

Diabetes mellitus is a chronic metabolic condition characterized by insufficient insulin production and reduced cellular sensitivity to insulin, leading to elevated blood glucose levels and disruptions in carbohydrate, protein, and fat metabolism. Type 2 diabetes (T2DM), the most prevalent form, is influenced by factors such as obesity, sedentary lifestyles, and westernized diets [2]. These risk factors, in turn, are related to an increase in circulating high glucose and high lipids (HG-HL) and lipopolysaccharide (LPS) [3,4]. Factors like chronic exposure to high glucose and high lipids (HG-HL), proinflammatory cytokines, ER stress, oxidative stress, epigenetic modifications, and dysregulated signaling pathways contribute to β-cell loss, dysfunctionality, and dedifferentiation. The latter results in a decrease in the expression of key transcription factors required for maintaining β-cell functionality and a concurrent elevation in factors specific to progenitor cells, triggering disruptions in insulin secretion [5,6]. Some of these transcription factors include: (1) β-cell specific transcription factors such as pancreatic and duodenal homeobox 1 (PDX1), v-maf musculoaponeurotic fibrosarcoma oncogene homolog A (MafA), neurogenic differentiation 1 (NEUROD1), NK6 homeobox 1 (Nkx6.1), and forkhead box protein O1 (FOXO1); (2) transcription factors specific to progenitor cells, such as Nanog and POU domain, class 5, transcription factor 1 (Pou5f1) [7,8,9,10].

Apart from the direct detrimental effects of the HG-HL condition and elevated LPS levels on β-cells, the inflammatory reactions, notably the imbalanced synthesis of IL-1β by infiltrating macrophages residing in the pancreas, significantly contribute to the loss and dysfunctionality of β-cells [11].

The generation of mature IL-1β is orchestrated through the activation of the NLRP3 inflammasome, a member of the NOD-like receptor (NLR) family that identifies pathogen-associated molecular patterns (PAMPs) and damage-associated molecular patterns (DAMPs) in the cytoplasm [12,13]. The activation of the NLRP3 inflammasome involves a biphasic process: priming (Signal 1) and assembly (Signal 2). Priming is typically initiated by TLR4 signaling, inducing NF-κB activation and subsequent transcriptional upregulation of NLRP3 and IL-1β genes [14]. Signal 2 is initiated by various PAMPs and DAMPs, leading to inflammasome assembly [12,15,16]. Caspase-1 activation by the NLRP3 inflammasome cleaves pro-IL-1β, pro-IL-18, and GSDMD. GSDMD cleavage forms pores in the plasma membrane, inducing cell death and releasing mature IL-1β and IL-18. This process plays a crucial role in various inflammatory conditions [17].

The combined effect of HG-HL conditions and exposure to LPS can lead to an overactive NLRP3 inflammasome, resulting in an exaggerated inflammatory response. This excessive activation is particularly relevant in the context of obesity, where chronic low-grade inflammation is often observed due to dysregulated metabolic processes and increased LPS levels associated with changes in gut microbiota composition and intestinal permeability [3,4,18].

The effects of eugenol on various facets of T2DM, including hyperglycemia, glycosylated hemoglobin (HbA1c), insulin levels, insulin resistance (IR), the levels of proinflammatory cytokines such as TNF-α and IL-6 [19], advanced glycation end products (AGEs) [20], and nephropathy [21] have been already studied. However, the inhibitory effects of eugenol on NLRP3 activation and also its impact on the impaired response of crucial genes/proteins involved in the dedifferentiation of pancreatic β-cells is not yet documented.

## 2. Results

### 2.1. Impact of Eugenol on the Viability of LPS-Induced THP-1 Monocyte/Macrophage Cells

The viability of THP-1 macrophages in response to three different eugenol doses was evaluated using the MTT assay. According to the results, none of the tested eugenol concentrations had a detrimental effect on LA-induced macrophage viability (Figure 1).

### 2.2. Eugenol Decreases IL-6, TNF-α, and COX-2 Levels in LPS-Induced THP-1 Macrophages

In the present study, we investigated the impact of three different concentrations of eugenol on the response of IL-6, mature TNF-α, and COX-2 levels in both LPS-induced and uninduced THP-1 macrophages using Western blot analysis. Our findings demonstrate that 15 μM eugenol effectively decreases the levels of COX-2 and IL-6 proteins in LPS-induced THP-1 macrophages (Figure 2). We also noted that 5 μM eugenol slightly increases the levels of COX-2 and IL-6 proteins. Furthermore, all three concentrations of eugenol were found to mitigate the level of mature TNF-α (Figure 2), while profoundly increasing the level of pro-TNF-α.

### 2.3. Eugenol Inhibits NLRP3 Inflammasome

The generation of mature IL-1β requires the activation of both stages of the NLRP3 inflammasome. The initial step involves the upregulation of NLRP3 and pro-IL-1β, while the second phase is triggered by the assembly of the NLRP3 inflammasome, leading to the cleavage of pro-caspase-1 and the subsequent production of mature IL-1β. Our Western blot results indicate that 10 μM and 15 μM eugenol mitigates the levels of both the immature and mature form of IL-1β in LA-induced THP-1 macrophages (Figure 3), while 5 μM eugenol increases their levels. Furthermore, 5 μM and 10 μM eugenol decreases the elevated level of NLRP3 in LA-induced THP-1 macrophages (Figure 3). We also noted that 15 μM eugenol further elevates the level of NLRP3.

### 2.4. All Three Doses of Eugenol Attenuate the Activation of NF-κB

NF-κB is a protein complex that controls transcription of genes encoding cytokines such as IL-1β, IL-6, and TNF-α [22]. It also regulates the expression of COX-2 [23]. Based on our Western blot findings, all three eugenol doses result in the downregulation of NF-κB phosphorylation and its subsequent translocation to the nucleus (Figure 4). Interestingly, eugenol did not exhibit any mitigative effects on the overall level of total NF-κB. This suggests that the inhibitory effects of eugenol on NF-κB activation are likely exerted through the suppression of its phosphorylation, thereby preventing the translocation of NF-κB into the nucleus.

### 2.5. Eugenol Inhibits the Cleavage of the C-Terminal Tale of Pannexin-1 (PANX-1)

PANX-1 is a ubiquitously expressed protein that forms large channels with high conductivity. PANX-1 has been found to play a crucial role in various inflammatory and injury responses. Accordingly, these channels contribute to increasing extracellular ATP, activating inflammasomes, and releasing proinflammatory cytokines [24]. Based on our findings, it has been observed that both 10 μM and 15 μM eugenol effectively inhibit the cleavage of the C-terminal tale of PANX-1 (cleaved PANX-1) in LPS-induced macrophages (Figure 5). This inhibition leads to the suppression of channel opening, subsequently reducing extracellular ATP levels (see Figure 5).

### 2.6. Eugenol Does Not Inhibit the Activation of STAT3

The IL-6/JAK/STAT3 pathway is a significant inflammatory signaling pathway. It plays a crucial role in numerous physiological and pathological processes, including immune regulation, angiogenesis, cell proliferation and differentiation. TYK2, a member of the Janus kinase (JAK) family of proteins, is known to phosphorylate and activate STAT3, leading to its subsequent nuclear translocation and transcriptional activity. Therefore, IL-6/TYK2 acts upstream of STAT3, contributing to the activation and regulation of STAT3-mediated cellular responses [25]. Based on our results, it is evident that although eugenol has an inhibitory effect on the phosphor-TYK2 (P-TYK2), it does not reduce the phosphorylation and subsequent activation of STAT3 (Figure 6).

### 2.7. Eugenol Mitigates Transcription of Proinflammatory Genes

To investigate the potential relationship between the anti-inflammatory effects of eugenol and the inhibition of NF-κB phosphorylation, we examined the transcription of various proinflammatory genes in LPS-induced THP-1 macrophages in response to eugenol. Our results revealed that both 10 μM and 15 μM eugenol downregulated the transcription of *IL-1β* in LPS-induced THP-1 macrophages, while 5 μM eugenol did not significantly impact the mRNA level of this gene. Additionally, 10 μM and 15 μM eugenol demonstrated downregulation of *IL-6* transcript. Surprisingly, eugenol exhibited a stimulatory effect on the transcription of *TNF-α* in LPS-induced THP-1 macrophages. Notably, only the highest dose of eugenol (15 μM) was able to downregulate the level of COX-2 transcript in THP-1 macrophages stimulated by LPS.

Moreover, all three doses of eugenol studied in this research exhibited inhibitory effects on the transcription of *NLRP3*, *PANX-1*, *ADAR-1*, and *Pro-Caspase-1* in LPS-induced THP-1 macrophages. Interestingly, like *TNF-α*, eugenol increased transcription of P2X7 in LPS-induced THP-1 macrophages (Figure 7).

### 2.8. Eugenol Lowers the Secreted IL-1β by LA-Induced THP-1 Macrophages

To establish a correlation between the Western blot analysis results for the mature form of IL-1β and the secretion of IL-1β in the medium, an ELISA assay was conducted. Consistent with the impact observed on the mature form of IL-1β, all three doses of eugenol led to a profound reduction in the elevated level of active IL-1β secreted in the medium by LA-induced THP-1 macrophages (Figure 8).

### 2.9. Eugenol Alleviates HG-HL-Induced β-Cell Loss, Likely through the Inhibition of Apoptosis

In order to investigate whether eugenol can prevent or decrease the HG-HL-induced β-cell loss, an MTT assay was employed using four treatments: control + ethanol (Ct + Et-OH), control + eugenol 10 μM (Ct + EUG 10), HG-HL, and HG-HL + eugenol 10 μM (HG-HL + Eug10). The administration of 10 μM eugenol resulted in a notable enhancement in cell viability in β-cells under HG-HL conditions (Figure 9). Our findings demonstrate that 10 μM eugenol mitigates HG-HL-induced β-cell loss for a duration of 6 days. Interestingly, 10 μM eugenol decreased the viability of β-cells in the Ct + EUG10 group as compared to Ct + Et-OH (Figure 9). It is noteworthy that the initial cell viability in the 96-well plate was uniform, indicating that an equal quantity of cells was introduced into each well on day 0.

Our Western blot analysis showed that 10 μM eugenol was able to reduce the elevated levels of pro-caspase-3, cleaved caspase-3 (C-caspase-3), pro-caspase-7, and cleaved PARP (C-PARP) in β-cells induced by HG-HL conditions (Figure 10), indicating its potential to mitigate apoptosis, which may play a crucial role in preventing β-cell loss.

### 2.10. Eugenol Does Not Improve HG-HL-Induced Impaired GSIS

GSIS is a vital physiological process orchestrated by the pancreatic β-cells to regulate blood glucose levels. However, when the body experiences elevated levels of both glucose and lipids, such as in conditions like hyperglycemia and hyperlipidemia, GSIS can become impaired. This impairment plays a significant role in the development and progression of T2DM. In the present study, we aimed to examine the potential of eugenol in ameliorating impaired GSIS induced by HG-HL conditions. To investigate this, pancreatic β-cells were pretreated with a concentration of 10 μM eugenol prior to exposure to HG-HL conditions for a duration of 48 h. Subsequently, the cells were washed with 1X HBSS and further incubated in HBSS containing 2.5 μM glucose (normal condition) or 16.5 μM glucose (high condition) for a period of 2 h, followed by collecting the supernatant to measure the levels of insulin secretion. Our findings reveal that eugenol does not exhibit a significant improvement in the impaired GSIS induced by HG-HL conditions (Figure 11).

### 2.11. Eugenol Increases the Levels of PDX-1 Protein and FOXO1 Phosphorylation in HG-HL-Induced β-Cells

PDX-1 is a critical transcription factor that plays a pivotal role in the development and function of pancreatic β-cells. It is primarily expressed in the pancreas, specifically in the islets of Langerhans, where β-cells are located. It regulates the expression of genes involved in β-cell development and function, including insulin itself. PDX-1 acts as a master regulator, coordinating the expression of multiple genes necessary for β-cell development and maintaining their mature phenotype [26]. Our results indicate that 10 μM and 15 μM eugenol increases the level of this transcription factor in HG-HL-induced β-cells (Figure 12).

FOXO1 (forkhead box O1) is another transcription factor that is highly expressed in various tissues, including pancreatic β-cells. It plays a significant role in regulating the function and survival of β-cells. Phosphorylation of FOXO1 is a critical post-translational modification that regulates its activity and subcellular localization. The phosphorylation of FOXO1 at multiple sites leads to its cytoplasmic retention and inhibition of its transcriptional activity [27]. According to our results, all three doses of eugenol increase the level of phospho-FOXO1 (P-FOXO1) in HG-HL-induced β-cells (Figure 12).

Thioredoxin-interacting protein (TXNIP), alternatively referred to as thioredoxin-binding protein-2, serves as a natural suppressor of thioredoxin (Trx) by diminishing its capability, thereby resulting in heightened levels of oxidative stress within the cells [28]. Based on the Western blot analysis findings, it was observed that only the lowest concentration of eugenol demonstrated the ability to decrease the protein expression level in β-cells subjected to HG-HL conditions (Figure 12).

### 2.12. Eugenol Partially Restores the Reduced Transcription of NEUROD, FOXO1, and SLC2A2 in HG-HL-Induced β-Cells

In order to examine the effect of eugenol on β-cell dedifferentiation induced by HG-HL conditions, the impact of 10 μM eugenol on the expression of the key biomarkers was investigated. The qRT-PCR analysis consisted of four experimental treatments: Ct + Et-OH, Ct + EUG 10, HG-HL, and HG-HL + EUG 10. According to our results, the administration of eugenol did not affect the expression levels of *Ins1*, *Ins2*, *PDX-1*, and *MafA* transcripts in β-cells exposed to HG-HL conditions (Figure 13). However, it significantly increased the expression of *FOXO1*, *NEUROD1*, and *SLC2A2* in HG-HL-induced β-cells (Figure 13).

## 3. Discussion

In this study, we aimed to investigate the anti-inflammatory properties and the impact of eugenol on HG-HL-induced β-cell loss and dedifferentiation.

### 3.1. The Anti-Inflammatory Effect of Eugenol on LPS/LA-Induced THP-1 Macrophages

In the current study, eugenol was utilized within the range of 5–15 μM, which falls within the safe dosage range suitable for long-term treatment. Carvalho et al. (2022) conducted a study on the effects of eugenol administration in rats. They found that doses of 20 and 40 mg/kg of eugenol resulted in a reduction in the activity of the Na^+^/K^+^ ATPase pump and blood glucose levels. These doses also led to increased hepatic glycogen content, superoxide dismutase activity, ferric reducing antioxidant power, as well as elevated levels of nitric oxide and malondialdehyde. However, the study observed that doses exceeding 20 mg/kg caused both structural and functional damage to the liver tissue in eugenol-treated rats. Therefore, the researchers concluded that a dose of 10 mg/kg of eugenol is considered safe for long-term treatment in rats, while higher doses can potentially harm the liver and impair its essential functions [29]; considering the rat weighs ~200 g, the rat received about 2 mg eugenol. Although our experiments were performed in vitro, we are well within this dose, as 10 μM eugenol is an equivalent of 1.64 mg/mL.

According to our results, eugenol downregulates pro-IL-1β, COX-2, and IL-6 in LPS/LA-induced THP-1 macrophages (Figure 2 and Figure 3), which may be mediated through the inhibition of NF-κB phosphorylation (Figure 4). This is further supported by the results of qRT-PCR with *IL-1β*, *COX-2*, and *IL-6* (Figure 7). Since the levels of *COX-2* mRNA corresponded to the levels of COX-2 protein, only the highest dose of eugenol inhibited *COX-2* mRNA and protein levels in LPS-induced macrophages; eugenol might exert its regulatory effect on COX-2 level in LPS-induced macrophages by reducing its transcription.

As our findings suggest, the impact of eugenol on the production of IL-6, pro-IL-1β, IL-1β, and COX-2 is contingent upon the dosage of eugenol administered. In our experiments, the application of 5 µM eugenol resulted in elevated levels of IL-6, pro-IL-1β, IL-1β, and COX-2 proteins. This observation can be explained by a phenomenon known as hormesis. Hormesis is a dose–response relationship where a substance, in this case, eugenol, can have different effects at low and high concentrations [30]. At low concentrations, eugenol might stimulate the immune system, leading to an increase in proinflammatory cytokines/proteins as a part of an initial defense response. However, at higher concentrations, eugenol might exert inhibitory effects on these cytokines, possibly by downregulating inflammatory cytokines/proteins. The exact mechanisms involved may vary depending on the specific cytokines and cells.

Although eugenol showed inhibitory effects on the level of mature TNF-α, surprisingly, it stimulated the level of both *TNF-α* mRNA and pro-TNF-α protein levels (Figure 2 and Figure 6), indicating that eugenol may exert its inhibitory impact on the response of mature TNF-α through the regulation of its post-translation phase. In this line, it should be mentioned that TNF-α is initially produced as an inactive precursor protein known as pro-TNF-α or TNF-α precursor, which remains bound to the cell membrane. The active and mature form of TNF-α (17 kDa) is produced when the pro-TNF-α undergoes proteolytic cleavage by the TNF-α converting enzyme (TACE), also referred to as ADAM17 [31]. The contrasting responses of pro-TNF-α and TNF-α proteins to eugenol may indicate that eugenol inhibits mature TNF-α levels by modulating TACE activity.

In line with our findings, the introduction of 60 μg/mL eugenol effectively reduced the transcriptional activity of IL-6 and inhibited the activation of NF-κB in LPS-induced THP-1 monocyte/macrophages. Additionally, this eugenol concentration led to a decrease in TNF-α secretion in THP-1 cells stimulated with LPS for 3 h [32]. Similarly, another study found that 200 μg/mL eugenol reduced the transcription and secretion of IL-1β and IL-6 in THP-1 cells treated with *Propionibacterium acnes* by inhibiting NF-κB activation [33]. However, unlike our findings, this study has shown that eugenol can reduce the levels of TNF-α transcript. The observed disparity can be ascribed to several contributing factors, including variations in eugenol dosages. Our study concentrated on investigating the anti-inflammatory properties of eugenol within the dosage range of 5–15 µM. In contrast, the referenced study employed a lower eugenol dosage. Additionally, distinctions in the choice of inflammatory inducers are notable; we utilized LPS/LA for inflammation induction, while they employed bacterial incubation. Furthermore, the discrepancy may also stem from differences in the duration of cellular incubation. Our study exposed cells to LPS for a duration of 4 h, whereas the other study entailed a more prolonged 24-h incubation period with the bacterium. In addition to the in vitro studies mentioned, two in vivo studies [34,35] have also demonstrated eugenol’s ability to inhibit the transcription of *IL-6*, *TNF-α*, and *COX-2* genes. However, these in vivo studies employed higher concentrations of eugenol compared to our current study.

TYK2 is a member of the JAK family, primarily responsible for phosphorylating STATs transcription factors. Furthermore, research conducted by Yang et al. in 2005 revealed that interferon-α triggers NF-κB activation through a pathway reliant on TYK2, implying that the phosphorylation of TYK2 might also play a direct role in NF-κB activation [36]. According to our research, eugenol exhibits inhibitory effects on both the IL-6/TYK2 axis (Figure 2 and Figure 6b) and NF-κB activation (Figure 4), suggesting that eugenol’s ability to mitigate NF-κB activation is partially attributed to its inhibition of the IL-6/TYK2 pathway.

In spite of the inhibitory impact of eugenol on IL6/TYK2, it did not mitigate the phosphorylation of STAT3 in LPS-induced macrophages (Figure 6a). Given that STAT3 plays a role in the production of certain proinflammatory cytokines, including IL-23 and IL-17, the impact of eugenol on the production of these proinflammatory cytokines should be tested to provide additional support for our results.

The production of the mature form of IL-1β is mediated by the assembly and subsequent activation of NLRP3 inflammasome. Our results revealed that eugenol downregulated the expression of *NLRP3* and *Pro-Caspase-1* genes (Figure 7), mediated through the mitigation of NFκB phosphorylation, and the level of NLRP3 protein in LPS-induced macrophages (Figure 3). When we consider these results in conjunction with the Western blot results for pro-IL-1β and mature IL-1β, as well as the ELISA results for secreted IL-1β, it strongly suggests that eugenol may have the capability to inhibit the assembly and subsequent activation of the NLRP3 inflammasome.

PANX-1 functions as a transmembrane transporter and plays a crucial role in increasing the levels of proinflammatory cytokines, particularly IL-1β. When macrophages are stimulated by different triggers, they enhance the expression and activation of PANX-1, leading to the release of internal molecules, including ATP. The released ATP then binds to purinergic receptors, specifically P2X7 receptors, found on macrophages. This binding event triggers a series of signaling pathways that ultimately activate the NLRP3 inflammasome, promote the maturation of caspase-1, and convert pro-IL-1β into its active form, IL-1β. Under normal circumstances, the PANX-1 channel is blocked by its cytoplasmic C-terminal region. However, in the presence of appropriate stimuli, caspases can cleave the C-terminus of PANX-1, causing the PANX-1 channels to open and release ATP [37].

Remarkably, eugenol has been discovered to reduce the expression of *PANX-1* (Figure 7). This finding suggests that eugenol may target PANX-1 as a potential mechanism through which it exerts its effects in mitigating the activation of the NLRP3 inflammasome. Furthermore, eugenol has demonstrated inhibitory properties in relation to PANX-1 cleavage, channel opening, extracellular ATP reduction, suppression of P2X7 activity, and the subsequent decrease in intracellular K^+^ levels. As a result, it inhibits the activation of the NLRP3 inflammasome. It is worth noting that the cleavage and subsequent opening of PANX-1 are mediated by the activity of caspase-3 and caspase-7 [38]. The inhibitory impact of eugenol on the mitigation of these apoptotic biomarkers in HG-HL-induced β-cells (Figure 10) implies that eugenol may target them as a potential mechanism to inhibit PANX-1 cleavage.

ADAR1, an enzyme called adenosine deaminase acting on RNA, plays a role in converting adenosine to inosine in both precursor and mature mRNA molecules. This RNA editing process facilitated by ADAR1 is responsible for generating a wide range of mRNA and protein isoforms. The expression of *ADAR1* has been observed in diverse cell types, suggesting its presence throughout various tissues [39].

The precise roles of ADAR1 in inflammation have been the subject of extensive investigation by various research groups. However, there is still an ongoing debate regarding its exact functions in this context. Certain studies propose that ADAR1 may exhibit anti-inflammatory properties, which are supported by a limited number of reports [40,41]. Conversely, the majority of published studies consistently demonstrate that ADAR1 actually enhances the inflammatory response, as evidenced by a substantial body of evidence [39,42,43].

Here, we showed that eugenol, at least in part, exerts its anti-inflammatory impact through the downregulation of this enzyme (Figure 7). This implies that eugenol may partially exert its anti-inflammatory effects through the modulation of RNA editing. This discovery could pave the way for further investigation into the role of eugenol in the modulation of RNA editing.

### 3.2. The Effects of Eugenol on HG-HL-Induced β-Cell Loss and Dedifferentiation

Using an in vitro system, we studied the effects of eugenol on the mitigation of HG-HL-induced β-cell loss and dedifferentiation and impaired GSIS. Our MTT results revealed that eugenol can decrease HG-HL-induced β-cell loss for a duration of 6 days (Figure 9), which is probably mediated through the mitigation of apoptosis as the inhibitory impact of eugenol on the levels of the key apoptotic biomarkers such as C-caspase-3, pro-caspase-3 and -7, and C-PARP has been documented (Figure 10). Consistently, the efficacy of eugenol in safeguarding islet of Langerhans cells and the liver and enhancing fasting blood glucose levels in diabetic rats has been demonstrated [44].

Our findings suggest that eugenol did not exhibit a statistically significant protective effect on impaired GSIS induced by HG-HL conditions (Figure 11). This outcome contrasts with a prior study that examined the influence of eugenol on insulin secretion in pancreatic islets from non-diabetic male mice [45]. However, it is important to note that our study specifically focused on evaluating the impact of eugenol on impaired GSIS induced by HG-HL conditions.

PDX-1 plays a crucial role in preserving the integrity of β-cells, and a decrease in its expression leads to the dedifferentiation of these cells. Conversely, inducing the expression of PDX-1 supports the maturity and functionality of β-cells. The regulation of the PDX-1 gene expression is a multifaceted process influenced by various factors such as nutrient substances, hormones, oxidative stress, and cytokines [46].

Based on our findings, the administration of all three eugenol doses resulted in the restoration of PDX-1 protein levels in HG-HL-induced β-cells (Figure 12). However, eugenol did not exhibit any significant effects on the levels of *PDX-1* mRNA (Figure 13).

The phosphorylation of FOXO1 at multiple sites, particularly at Ser256, diminishes its transcriptional activity, thereby decreasing its inhibitory effects on the transcription of *PDX-1* [27,47]. Additionally, FOXO1 and PDX-1 have distinct subcellular localizations within β-cells, indicating that the translocation of FOXO1 may influence the movement of PDX-1 between the nucleus and cytoplasm under oxidative stress conditions [48]. According to our findings, the administration of eugenol led to an elevation in the phosphorylation of FOXO1, consequently resulting in an increase in PDX-1 activity and translocation into the nucleus (12).

NEUROD1, a transcription factor, plays a critical role in regulating the function of β-cells. It directly activates genes that are involved in the maturation and proper functioning of β-cells [49]. According to our qPCR results, 10 μM eugenol could restore the level of *NEUROD1* mRNA in HG-HL-induced β-cells, providing extra support regarding the protective effects of eugenol against HG-HL-induced β-cell dedifferentiation and dysfunctionality (Figure 13).

The *SLC2A2* gene encodes the GLUT2 glucose transporter, which belongs to a family of transporters facilitating the entry of glucose into cells across the plasma membrane. GLUT2 is predominantly expressed in liver cells, pancreatic β-cells, intestines, and kidneys. It is essential for the optimal functioning of β-cells in rodents [50]. The administration of 10 µM eugenol effectively reversed the downregulation of *SLC2A2* mRNA in HG-HL-induced β-cells (Figure 13). This observation further strengthens the evidence supporting the protective effects of eugenol on β-cell function and maintenance.

Importantly, the administration of eugenol exhibited potential to ameliorate the reduced expression of *MafA*, *Ins1*, and *Ins2* in β-cells induced by HG-HL. However, these stimulatory effects were not supported by statistically significant responses to eugenol addition (Figure 13).

In summary, based on our results, it can be suggested that eugenol administration likely inhibits β-cell dedifferentiation by modulating certain critical genes and proteins that play a role in β-cell function and identity.

Considering the pivotal roles of MafA, PDX1, and NEUROD1 in β-cell function and maintenance, it is worth noting that the introduction of MafA, in combination with PDX1, NEUROD1, or NGN3, into adult pancreatic acinar cells or liver cells can prompt their conversion into insulin-producing cells [51,52]. Moreover, reduced expression of MafA has been identified as an early indicator of β-cell dysfunction in individuals with T2DM [53].

Previous studies have indicated that the primary source of upregulated proinflammatory cytokines, particularly IL-1β and TNF-α, in the pancreas under HG-HL conditions is the infiltrating macrophages. The dysregulation of TNF-α and IL-1β, produced by these infiltrating macrophages, can initiate the endoplasmic reticulum (ER) stress pathway in β-cells. This activation subsequently leads to β-cell apoptosis and the consequent loss of β-cells [54]. Both cytokines, originating from macrophages, bind to their receptors on β-cells, initiating intracellular signaling pathways that activate NF-κB transcription factor [55]. Activation of this transcription factor promotes the expression of the inducible nitric oxide (NO) synthase (iNOS) gene, resulting in increased production of nitric oxide (NO). It is important to note that NO plays a critical role in β-cell loss and dysfunction [54].

### 3.3. Potential Limitations and Future Directions

**In Vivo Validation:** This study primarily investigated eugenol’s anti-inflammatory and antidiabetic properties using in vitro models. To further validate and extend these findings, it is essential to conduct in vivo studies using animal models of inflammation and diabetes, providing a more comprehensive understanding of eugenol’s physiological effects.**Concentration Range Exploration:** The study used eugenol concentrations within a safe dosage range (5–15 µM) for long-term treatment. However, future research should explore a broader range of concentrations to assess potential dose-dependent effects on inflammation and β-cell function.**TRP Receptor Interaction:** To gain insights into the molecular mechanisms underlying eugenol’s effects, it is important to investigate its interaction with specific TRP receptors, particularly TRP1 and TRP8M. This research may uncover additional therapeutic targets for eugenol in inflammation and related conditions.**GSIS Mechanisms:** While eugenol showed promise in reducing inflammation and mitigating β-cell loss, it did not significantly protect against impaired GSIS (glucose-stimulated insulin secretion). Further studies are needed to elucidate the underlying mechanisms and identify specific conditions or factors that could enhance eugenol’s effectiveness in improving GSIS.**Long-Term Safety Profile:** Additional research is warranted to assess the long-term effects and safety profile of eugenol, especially regarding its potential use as a therapeutic agent for inflammation and diabetes. This includes evaluating potential side effects, drug interactions, and conducting thorough pharmacokinetic and pharmacodynamic studies.**NLRP3 Inflammasome Pathway:** Future work should explore the effects of eugenol on other key components of the NLRP3 inflammasome activation pathway. This should involve analyzing its impact on post-translational modifications and downstream signaling events, providing a more comprehensive understanding of eugenol’s anti-inflammatory mechanisms.

## 4. Materials and Methods

### 4.1. Chemicals, Reagents, and Cell Lines

L-Glutamine (TMS-002-C), sodium pyruvate (S8636), HEPES (TMS-003-C), β-mercaptoethanol (ES-007), lipopolysaccharide (LPS) derived from *Escherichia coli* O111:B4 (L4391), adenosine 5′-triphosphate (ATP) disodium salt hydrate (A6419), and INS-1 832/13 rat insulinoma cells (SCC207) were obtained from EMD Millipore Corporation (Temecula, CA, USA). Phorbol-12-myristate-13-acetate (PMA) was purchased from Enzo (BML-PE160-0005, Farmingdale, NY, USA).

Roswell Park Memorial Institute medium (RPMI 1640) (350-060-CL), RPMI 1640 (350-000-CL) culture media, and 1X phosphate-buffered saline (PBS) (311-010-CL) were obtained from Wisent Inc. (Saint-Jean-Baptiste, QC, Canada). The media were supplemented with 10% heat-inactivated premium-grade fetal bovine serum (FBS) ((10082147) acquired from Fisher Scientific Company, Ottawa, ON, Canada). The THP-1 monocyte/macrophage (ATCC TIB-202) cell line was obtained from the American Type Culture Collection (Rockville, MD, USA).

### 4.2. Cell Culture and Treatments

The THP-1 monocyte/macrophage cell line, with passages between 8 and 13, was utilized for inflammation-related experiments. Differentiation of non-adherent monocytes into adherent macrophages was achieved by treating them with 50 ng/mL PMA for 48 h. Subsequently, adherent macrophages were washed twice with PBS and allowed to recover for 24 h in PMA-free medium before exposure to various concentrations of eugenol for 1 h. To induce proinflammatory responses, the cells were then incubated with 500 ng/mL LPS for 4 h, followed by RNA extraction.

For Western blot and ELISA studies, the second phase of NLRP3 inflammasome activation was induced by incubating cells with 5 mM ATP for 30 min. The supernatant from the LPS + ATP (LA)-treated cells was collected for further analysis using ELISA, and whole cellular lysates were prepared from the treated cells for Western blot analysis.

For diabetes-related experiments, INS-1 832/13 rat insulinoma cells, maintained at passages 5 to 10, were cultured in RPMI 1640 medium (350-060-CL) enriched with specific components, including 2 mM L-Glutamine, 1 mM sodium pyruvate, 10 mM HEPES, 0.05 mM β-mercaptoethanol, 10% fetal bovine serum (FBS), and 2.5 mM glucose. Afterwards, cells were exposed to specific dose(s) of eugenol or an equivalent amount of the control vehicle (ethanol) for a period of 2 h. Subsequently, the cells were subjected to high glucose (HG) conditions, with a glucose concentration of 25 mM, and high lipid (HL) conditions, with 400 μM palmitic acid, for the following 48 h. Following these experimental treatments, RNA was isolated and whole cellular lysates were prepared from the treated cells to perform quantitative real-time polymerase chain reaction (qRT-PCR) and Western blot analysis, respectively.

### 4.3. MTT Assay

To assess the impact of eugenol on the viability of LA-induced THP-1 macrophages, we conducted an MTT assay. Non-adherent monocytes, previously incubated with 50 ng/mL PMA, were seeded into individual wells of a 96-well plate, with each well containing 1 × 10^6^ cells. Subsequently, macrophages were treated with various concentrations of eugenol or an equivalent volume of the vehicle for 1 h. Following this, the cells were incubated with a medium containing eugenol, the vehicle, and/or LPS + ATP for 24 h at +37 °C with a 5–6.5% CO_2_ concentration.

The study also employed an MTT assay to assess the effect of eugenol on the viability of β-cells under conditions of HG-HL exposure. The cells were initially seeded at a concentration of 5 × 10^5^ cells per well in 100 μL of culture medium, which included eugenol or an equivalent amount of ethanol (used as a control), for a period of 2 h. Subsequently, the cells were incubated with a medium containing eugenol, control, and/or HG-HL conditions for 48 h at a temperature of +37 °C and a carbon dioxide concentration of 5–6.5%.

After the incubation period, 10 μL (10% of each well content) of an MTT labeling reagent (3-(4,5-Dimethylthiazol-2-yl)-2,5-diphenyltetrazolium bromide) (11465007001) from Millipore Sigma Canada Ltd. (Oakville, ON, Canada) was added to each well. The microplate was then incubated for 4 h in a humidified environment at +37 °C with a carbon dioxide concentration of 5–6.5%. Following this, 100 μL of a solubilization solution was added to each well, and the plate was left to stand overnight in a humidified incubator at +37 °C with a carbon dioxide concentration of 5–6.5%. The absorbance at a wavelength of 595 nm was subsequently measured using a plate reader (FLUOstar Omega, BMG LABTECH, Offenburg, Germany).

### 4.4. Western Blotting Analysis

The treated cells were washed twice with cold PBS and then lysed using RIPA buffer. Following centrifugation at 13,000 rpm for 15 min, the resulting supernatant was collected and transferred to a new microtube. The protein content was determined using the Bradford assay, and these lysates were utilized for Western blotting. To separate the proteins, polyacrylamide gels with varying concentrations (8%, 10%, 12%, 15%) were employed, and subsequently, the proteins were transferred onto polyvinylidene difluoride (PVDF) membranes (specifically, Amersham Hybond^®^ P (RPN2020F), from GE Healthcare, Oakville, ON, Canada). Prior to incubation with primary antibodies overnight at 4 °C, the membranes were blocked using a PBS solution containing 1% tween 20 (PBST) and 5% milk. After three washes with PBST, the membranes were exposed to secondary antibodies for two hours at room temperature, followed by another round of PBST washes. For the detection of immunoreactivity, peroxidase-conjugated antibodies were used, and visualization was achieved using the ECL Plus Western Blotting Detection System. The densitometry of bands was measured and normalized with that of GAPDH or actin using ImageJ. Information about the primary antibodies is provided in Appendix A.

### 4.5. Quantitative Real-Time Polymerase Chain Reaction (qRT-PCR) Analysis

Total RNA was extracted from cellular material utilizing the TRIzol^®^ Reagent (15596018, Invitrogen, Life Technologies Inc., Burlington, ON, Canada), as per the manufacturer’s instruction. Following extraction, the concentration of RNA samples was measured using Nanodrop (Thermo Fisher Scientific, Waltham, MA, USA). A total of 1 μg of total RNA was employed to synthesize complementary DNA (cDNA), utilizing a specialized cDNA synthesis kit known as iScript™ Reverse Transcription Supermix (1708841), obtained from BioRad Laboratories (Saint-Laurent, QC, Canada). The synthesized cDNA was then employed as a template for qRT-PCR, with 1 μL of cDNA being utilized in each reaction. The qRT-PCR reactions were carried out using a SsAdvanced^TM^ Universal Inhibitor-Tolerant SYBR Green Supermix (1725017), purchased from Bio-Rad Laboratories (Saint-Laurent, QC, Canada). The primers required for the qRT-PCR analysis were designed employing online Integrated DNA Technologies (IDT), PrimerQuest Tool and purchased from Eurofins (Ottawa, ON, Canada) (Appendix A).

### 4.6. Glucose-Stimulated Insulin Secretion (GSIS) Assay

The experiment began by seeding β-cells in a 24-well plate, with each well initially containing 0.5 × 106 cells. These cells were then left to incubate for a period of 2 days. Following the incubation, the cells were subjected to a 2-h pretreatment using media containing either eugenol or a control vehicle (ethanol). After this pretreatment, the cells were exposed to HG-HL conditions for a total duration of 48 h.

To assess glucose-stimulated insulin secretion (GSIS), a specialized solution was prepared using HBSS with precise salt concentrations and the addition of bovine serum albumin, all adjusted to a pH of 7.2. The cells were then thoroughly washed with HBSS. During the second wash, which extended for 1 h, any residual substances were removed. The experimental groups were distributed across two wells. One well was treated with HBSS containing 10 μM eugenol and 2.5 mM glucose, representing fasting blood glucose in rats. In contrast, the other well received HBSS with 10 μM eugenol and 16.5 mM glucose, representing postprandial blood glucose. Both sets of wells were subsequently incubated for 2 h, and after this incubation period, the solutions were collected for analysis via insulin ELISA.

We utilized the Rat/Mouse Insulin ELISA kit (EZRMI-13K) from Sigma Aldrich (Oakville, ON, Canada) for our ELISA analysis. To prepare for the ELISA, we placed the ELISA strips into a holder, and each well underwent a triple rinse with 300 μL of diluted Wash Buffer. Following the rinsing, we introduced 10 μL of each sample into the wells, and this was succeeded by the addition of 80 μL of the Detection Antibody. The entire plate was sealed and then incubated at room temperature on an orbital microtiter plate shaker. After incubation, we removed the sealer and carefully emptied the solutions from the wells. The wells then experienced three additional washes with the diluted Wash Buffer and were gently tapped on a paper towel to eliminate any excess buffer. Next, we introduced 100 μL of Enzyme Solution into each well, resealed the plate, and incubated it with moderate shaking for 30 min. Once this step was completed, we performed six more washes with the diluted Wash Buffer. Subsequently, 100 μL of Substrate Solution was dispensed into each well. The plate was sealed once more and gently shaken for a period of 5–20 min. After removing the sealer, we added 100 μL of Stop Solution to each well and briefly shook the plate to ensure thorough mixing. Finally, we measured the absorbance of the plate at 450 nm and 590 nm using SpectraMax i3x Multi-Mode Microplate Reader (Molecular Devices, San Jose, CA, USA).

### 4.7. The IL-1β Enzyme-Linked Immunosorbent Assay (ELISA)

We assessed the release of IL-1β in response to various treatments using a Human IL-1β/IL-1F2 Quantikine ELISA Kit (DLB50) from R&D Systems, Inc. (Minneapolis, MN, USA). Each well in the ELISA plate was loaded with 200 μL of either a standard, control, or sample, and they were left to incubate at room temperature for a period of 2 h. Following this incubation, the wells underwent a triple washing procedure using a specialized washing buffer. Subsequently, 200 μL of human IL-1β conjugate was introduced to each well and covered with a fresh adhesive strip. Another round of incubation, lasting for 1 h at room temperature, was carried out. Afterward, we performed another aspiration and washing cycle. Then, 200 μL of Substrate Solution was added to each well, and the entire plate was incubated for 20 min at room temperature. To conclude the process, 50 μL of Stop Solution was added to each well, and we measured the optical density of the wells at 450 nm and 570 nm using a plate reader (SpectraMax i3x Multi-Mode Microplate Reader, Molecular Devices, San Jose, CA, USA). To account for any optical anomalies in the plate, we subtracted the readings at 570 nm from those at 450 nm.

It is important to note that due to the high concentration of secreted IL-1β in the medium, the samples were diluted by a factor of 50 before being subjected to the ELISA assay.

### 4.8. Statistics

The collected data were subjected to statistical analysis using one-way analysis of variance (ANOVA), followed by Tukey and Dunnett’s tests for comparing mean values. This analysis was performed using GraphPad Prism software, version 9.5.1 (733).

## 5. Conclusions

In this study, we demonstrated the ability of eugenol to reduce the production of proinflammatory cytokines, specifically IL-1β and TNF-α, in macrophages. This suggests that eugenol has the potential to mitigate the dysregulated production of these proinflammatory cytokines in infiltrating macrophages, thereby providing protection for β-cell survival and function. Furthermore, we observed that eugenol directly decreased the loss of β-cells induced by HG-HL. This effect of eugenol on β-cell loss may be attributed to its ability to mitigate apoptosis (Figure 14).

## Figures and Tables

**Figure 1 molecules-28-07619-f001:**
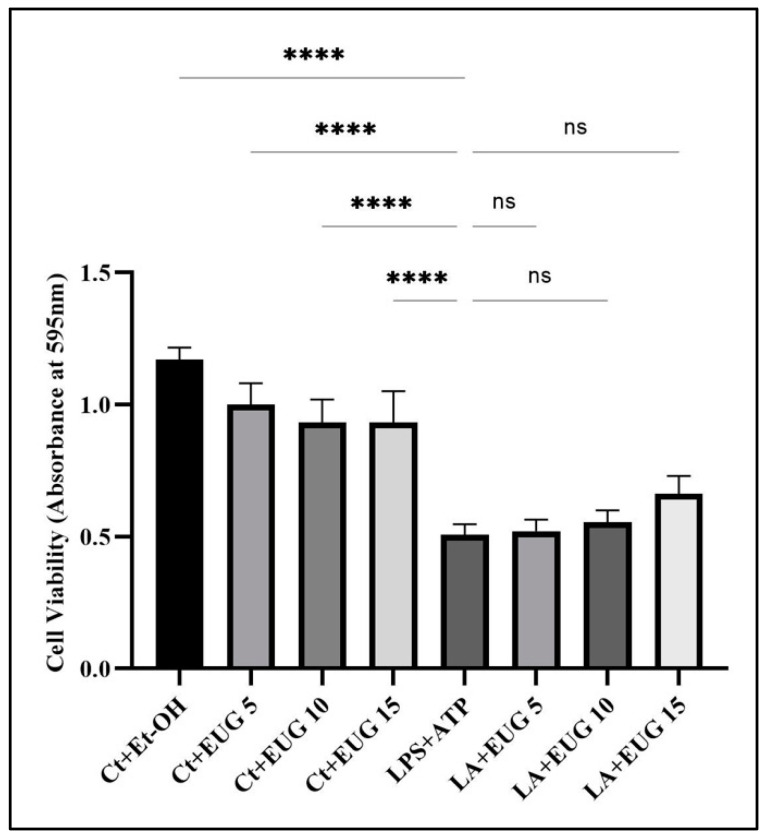
Impact of eugenol treatment on the viability of LA-induced macrophages. THP-1 macrophages were seeded in a 96-well plate and pretreated with three different doses of eugenol before incubation with LPS and ATP, as described elsewhere (see Methods). The treated cells were then incubated with MTT reagent for 4 h, followed by the addition of a resolving solution. The data are shown as the mean value with SD, n = 3. The abbreviations used in the figure are as follows: Ct + Et-OH (control + ethanol) and LA (LPS + ATP). The asterisks show significant difference: 4 asterisks—*p* < 0.0001; ns—nonsignificant.

**Figure 2 molecules-28-07619-f002:**
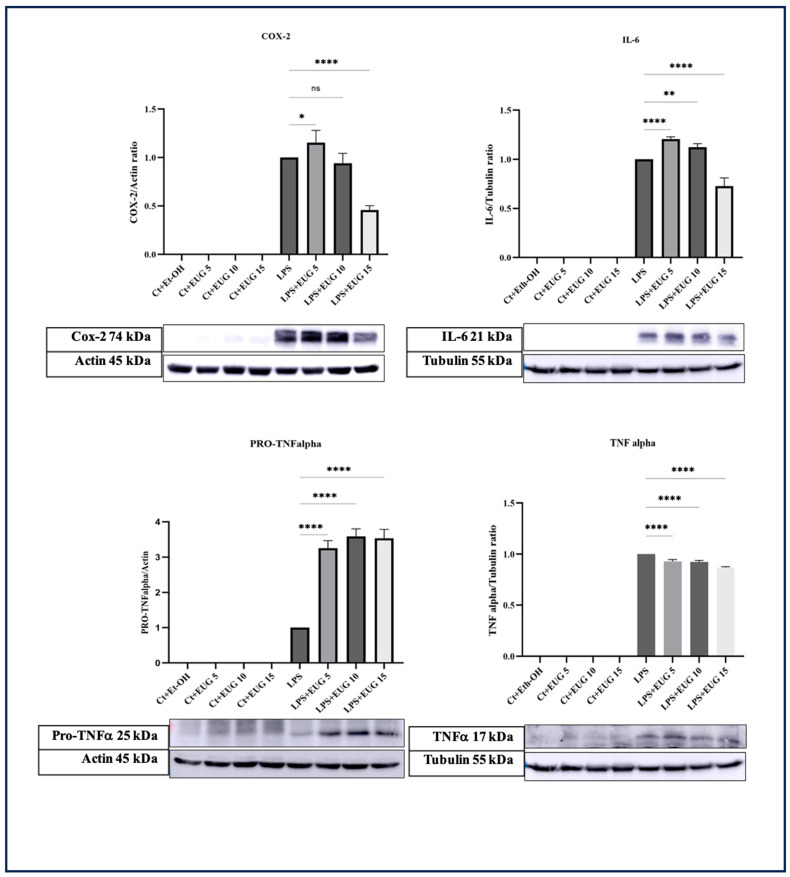
The Western blot analysis of IL-6, TNF-α, and COX-2 in THP-1 macrophages in response to eugenol. THP-1 macrophages pretreated with eugenol were exposed to LPS for 4 h, followed by cellular lysate preparation for Western blot analysis. Eugenol reduced the levels of IL-6 mature TNF-α. All data are presented as mean value +/− SD, n = 3 measurements. Abbreviation: Ct + Et-OH (control + ethanol), Ct + EUG (control + eugenol), LPS + EUG (LPS + eugenol). The asterisks show significant difference: one—*p* < 0.05; two—*p* < 0.01; four—*p* < 0.0001; ns—nonsignificant.

**Figure 3 molecules-28-07619-f003:**
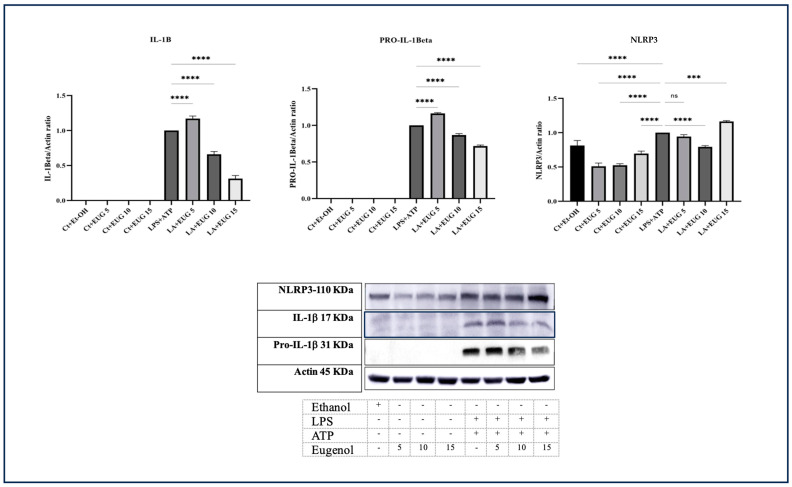
Western blot analysis of pro-IL-1β, mature IL-1β, and NLRP3 in response to eugenol in both LA-induced and uninduced THP-1 macrophages. The response of pro-IL-1β, mature IL-1β, and NLRP3 proteins to three doses of eugenol in both LA-induced and uninduced THP-1 macrophages was determined by Western blot analysis. All data are presented as mean value +/− SD, n = 3 measurements. Abbreviation: Ct + Et-OH (control + ethanol), Ct + EUG (control + eugenol), LPS + EUG (LPS + eugenol), LA (LPS + ATP). The asterisks show significant difference: three—*p* < 0.001, four—*p* < 0.0001; ns—nonsignificant.

**Figure 4 molecules-28-07619-f004:**
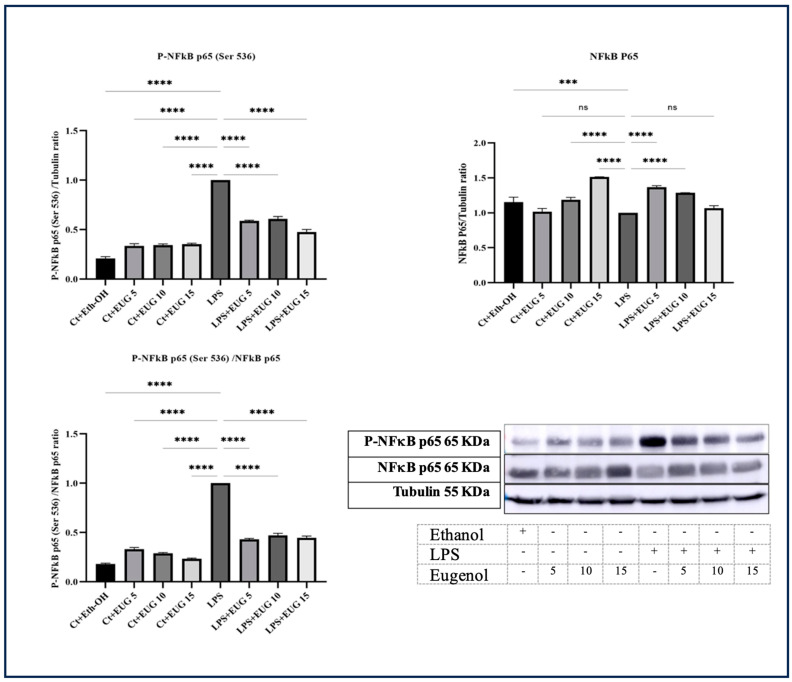
Western blot analysis of P-NF-κB P65 (Ser536), NF-κB P65, and P-NF-κB P65 (Ser536)/NFκB P65 in LA-induced and uninduced THP-1 macrophages treated with eugenol. Western blot analysis was performed to determine the impact of eugenol on the levels of P-NF-κB P65 (Ser536) and P-NF-κB P65 (Ser536)/NF-κB P65 in THP-1 macrophages induced by LPS. All data are presented as mean value ± SD, n = 3 measurements. Abbreviation: Ct + Et-OH (control + ethanol), Ct + EUG (control + eugenol), LPS + EUG (LPS + eugenol). The asterisks show significant difference, where three—*p* < 0.001; four—*p* < 0.0001; ns—nonsignificant.

**Figure 5 molecules-28-07619-f005:**
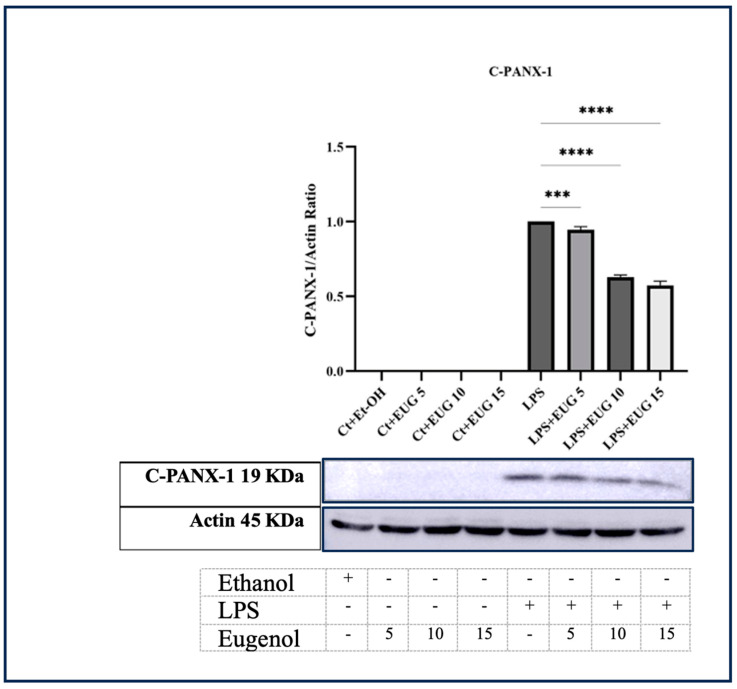
The response of cleaved PANX-1 (C-PANX-1) to three doses of eugenol in both LA-induced and uninduced THP-1 macrophages. Western blot analysis was performed to determine the effect of eugenol on the elevated level of C-PANX-1 (PANX-1 C-terminal) in THP-1 macrophages stimulated with LPS. All data are presented as mean value ± SD, n = 3 measurements. Abbreviation: Ct + Et-OH (control + ethanol), Ct + EUG (control + eugenol), LPS + EUG (LPS + eugenol). The asterisks show significant difference, where three—*p* < 0.001; four—*p* < 0.0001.

**Figure 6 molecules-28-07619-f006:**
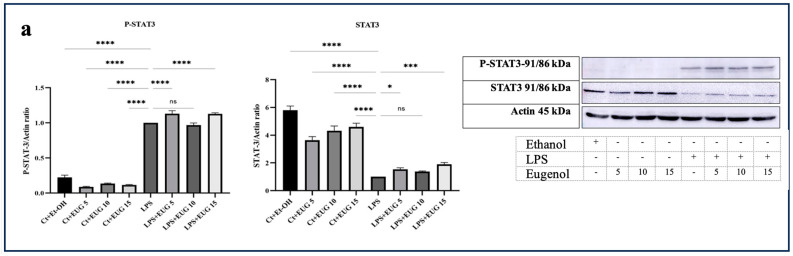
The Western blot analysis of P-STAT3, total STAT3, P-TYK2, and TYK2 in response to eugenol in both LA-induced and uninduced THP-1 macrophages. (**a**) The levels of P-STAT3 and total-STAT3 proteins in both LA-induced and uninduced THP-1 macrophages upon eugenol treatment. (**b**) The levels of P-TYK2 and TYK2 proteins in both LA-induced and uninduced THP-1 macrophages upon eugenol treatment. All data are presented as mean value ± SD. Abbreviation: Ct + Et-OH (control + ethanol), Ct + EUG (control + eugenol), LPS + EUG (LPS + eugenol). The asterisks show significant difference: one—*p* < 0.05; three—*p* < 0.001; four—*p* < 0.0001; ns—nonsignificant.

**Figure 7 molecules-28-07619-f007:**
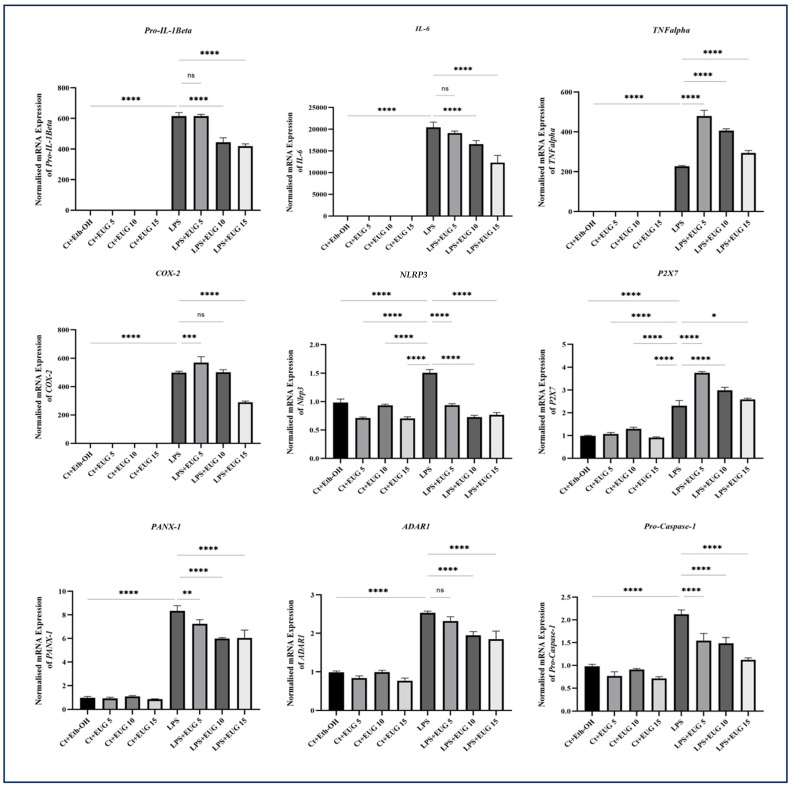
The qRT-PCR analysis of *IL-1β*, *IL-6*, *pro-TNFα*, *COX-2*, *NLRP3*, *PANX-1*, *P2X7*, *ADAR-1*, and *Pro-Caspase-1* in response to eugenol in both LPS-induced and uninduced THP-1 macrophages. The normalized mRNA expression levels of *IL-1β*, *IL-6*, *pro-TNFα*, *COX-2*, *NLRP3*, *PANX-1*, *P2X7*, *ADAR-1*, and *Pro-Caspase-1* were assessed in response to three doses of eugenol in both LPS-induced and uninduced THP-1 macrophages. *GAPDH* and *β-Actin* were used as control genes to normalize the transcript levels. All data are presented as mean value ± SD, n = 3. Abbreviations: Ct: control, Meth: methanol, EUG: eugenol, Et-OH: ethanol. The asterisks show significant difference: one—*p* < 0.05; two—*p* < 0.01; three—*p* < 0.001; four—*p* < 0.0001; ns—nonsignificant.

**Figure 8 molecules-28-07619-f008:**
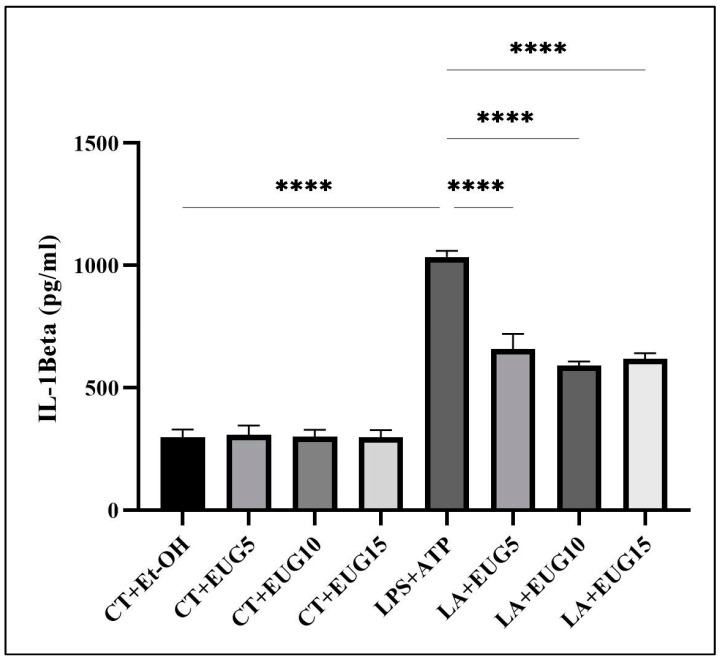
The effect of eugenol on IL-1β secretion in LA-induced and uninduced THP-1 macrophages. The impact of eugenol at doses of 5, 10, and 15 μM on the secretion of IL-1β in both LA-induced and uninduced THP-1 macrophages was measured using ELISA. All data are presented as mean value +/− SD, n = 3 measurements. Abbreviation: LA (LPS + ATP), Ct + Et-OH (control + ethanol), EUG (eugenol). The asterisks show significant difference, *p* < 0.0001.

**Figure 9 molecules-28-07619-f009:**
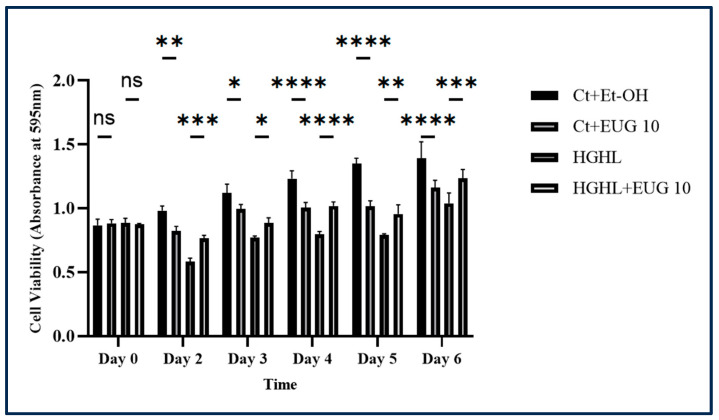
The effect of 10 μM eugenol on the viability of β-cells under HG-HL conditions for a duration of 6 days. INS-1 832/13 rat insulinoma cells were seeded in 96-well plates and then incubated with 10 μM eugenol and/or HG-HL conditions for specific time periods. After each time point, cells were treated with an MTT reagent for four hours, followed by incubation in solubilization buffer for the next 24 h. All data are presented as the mean value ± SD, n = 3 measurements. Abbreviations: Ct + Meth: control + methanol, HG-HL: high glucose-high lipid, Ct + Et-OH: control + ethanol. The asterisks show significant difference: one—*p* < 0.05; three—*p* < 0.001; two—*p* < 0.01; four—*p* < 0.0001; ns—nonsignificant.

**Figure 10 molecules-28-07619-f010:**
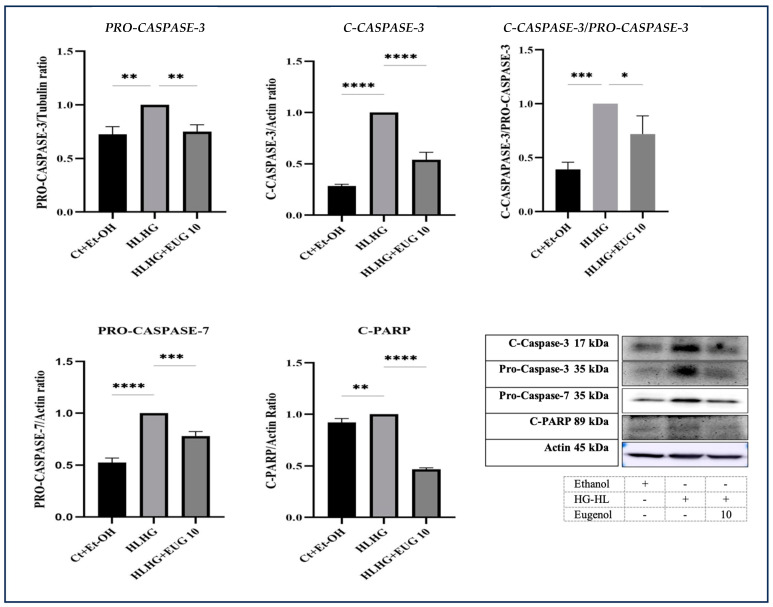
The Western blot analysis of some crucial apoptotic biomarkers in response to eugenol. INS-1 832/13 rat insulinoma cells were incubated with 10 μM eugenol and/or HG-HL conditions for 48 h, followed by cell lysate preparation for subsequent Western blot analysis. The response of pro-caspase-7, pro-caspase-3, C-caspase-3, C-PARP, and the C-caspase-3/pro-caspase-3 ratio to 10 μM eugenol in both HG-HL-induced and uninduced β-cells was determined using Western blot analysis. All data are presented as the mean value ± SD, n = 3 measurements. Abbreviations: Ct + Et-OH (control + ethanol), HG-HL (high glucose-high lipid), EUG (eugenol). The asterisks show significant difference: one—*p* < 0.05; two—*p* < 0.01; three—*p* < 0.001; four—*p* < 0.0001; ns—nonsignificant.

**Figure 11 molecules-28-07619-f011:**
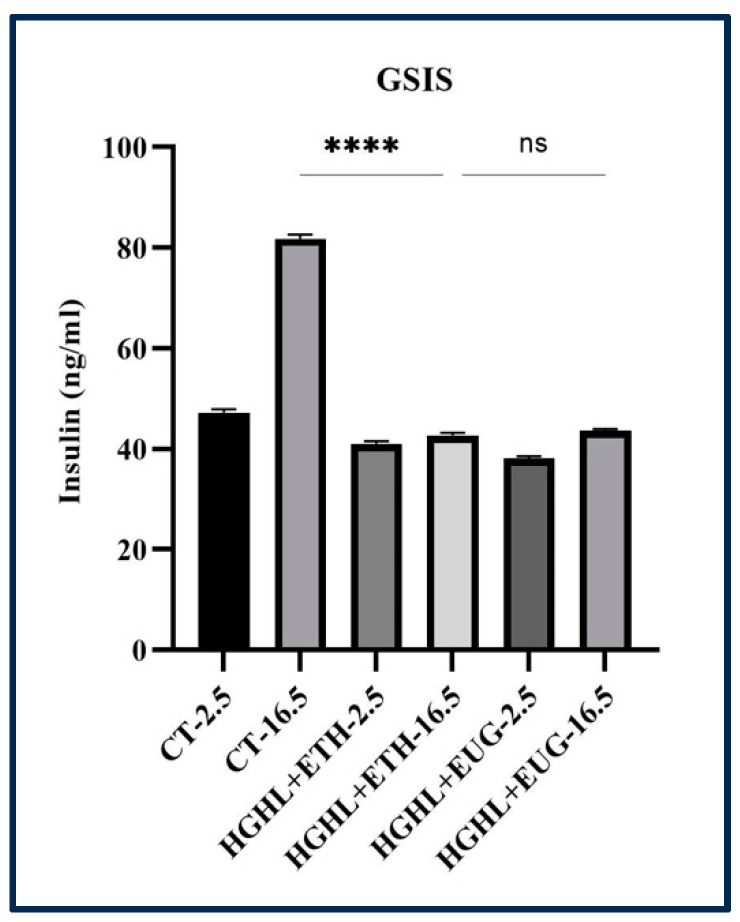
The results of GSIS assay conducted on HG-HL-induced β-cells treated with 10 μM eugenol. The GSIS response of HG-HL-induced β-cells to eugenol in medium containing 2.5 mM glucose (to simulate fasting blood glucose) and 16.5 mM glucose (to simulate postprandial blood glucose) was conducted as detailed in “Materials and Methods”. All data are presented as the mean value ± SD, n = 3 measurements. Abbreviations: Ct + Et-OH (control + ethanol) and HG-HL (high glucose-high lipid), EUG (eugenol). The asterisks show significant difference: four—*p* < 0.0001; ns—nonsignificant.

**Figure 12 molecules-28-07619-f012:**
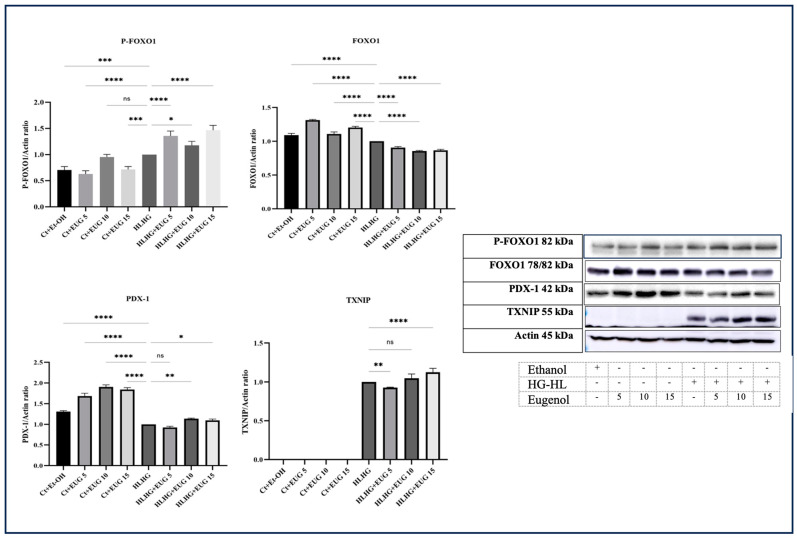
The Western blot analysis of P-FOXO1, FOXO1, PDX-1, and TXNIP in response to eugenol in both HG-HL-induced and uninduced β-cells. The Western blot analysis was conducted to determine the levels of P-FOX1, FOX1, PDX-1, and TXNIP in HG-HL-induced and uninduced β-cells in response to eugenol. All data are presented as the mean value ± SD, n = 3 measurements. Ct + Et-OH (control + ethanol) and HG-HL (high glucose-high lipid), EUG (eugenol). The asterisks show significant difference: one—*p* < 0.05; two—*p* < 0.01; three—*p* < 0.001; four—*p* < 0.0001; ns—nonsignificant.

**Figure 13 molecules-28-07619-f013:**
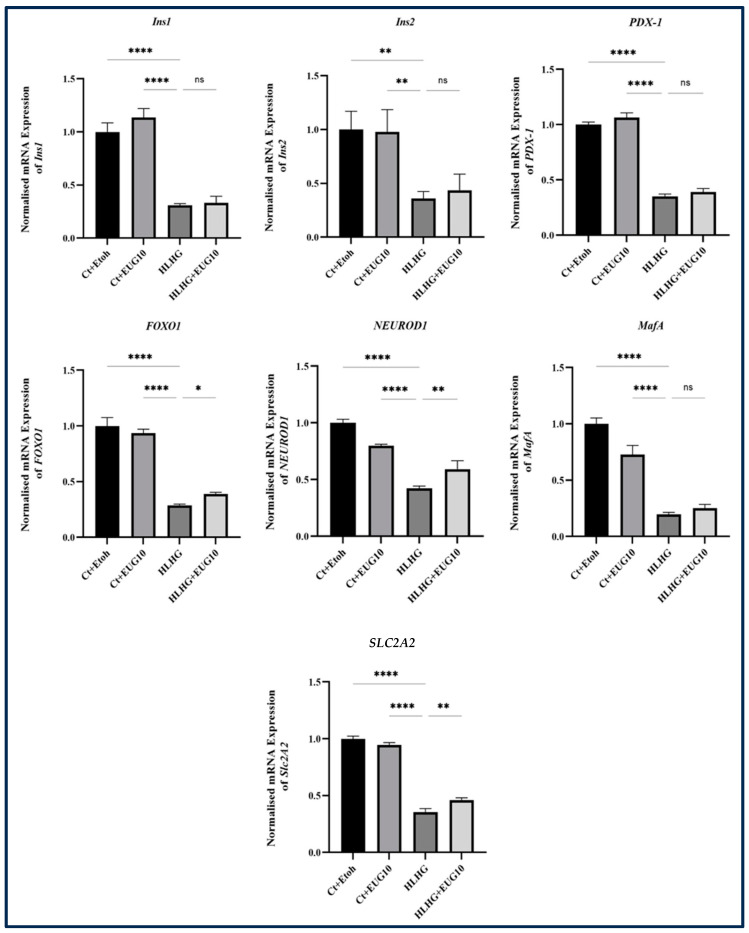
The response of key transcripts involved in β-cell dedifferentiation to 10 µM eugenol in both HG-HL induced and uninduced β-cells. The normalized mRNA levels of *Ins1*, *Ins2*, *PDX-1*, *FOXO1*, *NEUROD1*, *MafA*, and *SLC2A2* in both HG-HL-induced and uninduced β-cells treated with 10 µM eugenol were measured using qRT-PCR. *α-Tubulin* was used as control gene to normalize the transcript levels. All data are presented as the mean value ± SD, n = 3. Abbreviations: Ct + Et-OH (C=control + ethanol) and HG-HL (high glucose-high lipid), EUG (eugenol). The asterisks show significant difference: one—*p* < 0.05; two—*p* < 0.01; four—*p* < 0.0001; ns—nonsignificant.

**Figure 14 molecules-28-07619-f014:**
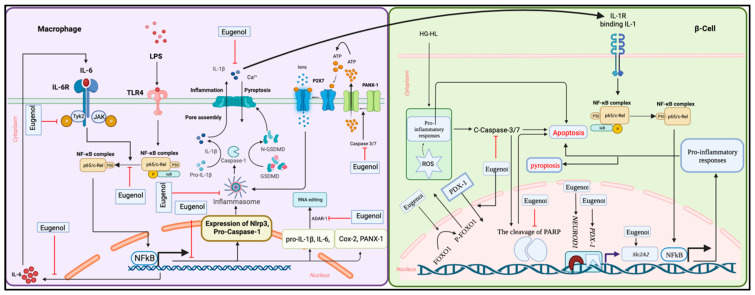
Eugenol mitigates β-cell loss by suppressing inflammatory responses in infiltrating pancreatic macrophages and β-cell apoptosis. Eugenol exerts its anti-inflammatory effects on macrophages stimulated by LPS by mitigating the activity of NFκB, resulting in the downregulation of proinflammatory cytokines/genes such as *IL-1β*, *IL-6*, *COX-2*, *NLRP3*, *pro-caspase-1*, and *PANX-1*. The mitigation of IL-6/P-TYK2 could be a significant factor in how eugenol inhibits the activation of NFκB. Notably, the inhibition of PANX-1 cleavage and opening by eugenol may play a crucial role in the suppressing effect of eugenol on the NLRP3 inflammasome, pro-caspase-1 activation, and the subsequent production and release of mature IL-1β. This inhibitory impact of eugenol on IL-1β secretion by infiltrating macrophages can indirectly reduce β-cell loss. Furthermore, eugenol exhibits potential in downregulating β-cell loss induced by high glucose and high lipid levels, potentially through the regulation of apoptosis biomarkers. Additionally, eugenol restores decreased levels of *NEUROD1* and *SLC2A2* transcripts and PDX-1 protein in β-cells exposed to high glucose and high lipid conditions, indicating its potential to counteract β-cell dedifferentiation. Eugenol’s inhibitory effects on β-cell dedifferentiation in high glucose and high lipid conditions also involve the stimulation of P-FOXO1, leading to the nuclear localization of PDX-1.

## Data Availability

Data are contained within the article and Appendix A.

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
