# Peer review of "Anti-Inflammatory Properties of Eugenol in Lipopolysaccharide-Induced Macrophages and Its Role in Preventing β-Cell Dedifferentiation and Loss Induced by High Glucose-High Lipid Conditions"

_molecules, 2023, doi:10.3390/molecules28227619_

Round 1

Reviewer 1 Report

Comments and Suggestions for Authors

The manuscript described the use of natural product eugenol for anti-inflammation and anti-diabetes. Basically, the in vitro data especially for western blot and PCR is sufficient. However, author should address the following questions before consideration for publication:

1) Figure 2 at 5 μM, the combination of EUG with LPS seemed to stimulate the expression of COX-2 and IL-6, this seemed to stimulate inflammation, the author should explain this phenomenon in the discussion

2) some details of Figures should be further refined. e.g. Figure 12 red line below KDa

3) Discussion is not sufficient. e.g. The present study is in vitro study and so in the discussion the author should talk about what kinds of animal models they can use to validate their conclusions 

Comments on the Quality of English Language

The quality of English Language in this manuscript is fine  

Author Response

  • Figure 2 at 5 μM, the combination of EUG with LPS seemed to stimulate the expression of COX-2 and IL-6, this seemed to stimulate inflammation, the author should explain this phenomenon in the discussion.

Response: An explanation was added to the discussion part.

  • some details of Figures should be further refined. e.g., Figure 12 red line below KDa.

Response: Figure 12 was modified to address the provided comment.

  • Discussion is not sufficient. e.g. The present study is in vitro study and so in the discussion the author should talk about what kinds of animal models they can use to validate their conclusions. 

Response: A segment has been incorporated into the discussion section to address these comments. Within this section, the limitations and potential future directions of the study have been delineated.

Reviewer 2 Report

Comments and Suggestions for Authors

The study is both interesting and relevant; however, its density makes it challenging to digest. The numerous assays can render the reading somewhat tedious and monotonous. It could easily be split into two separate papers. The discussion requires further elaboration, especially when comparing it to other studies. Please enhance the quality of the figures; adding color to the graphs would be highly beneficial. Additionally, elucidate on the future prospects based on the study's conclusions.

Author Response

  • The numerous assays can render the reading somewhat tedious and monotonous. It could easily be split into two separate papers.

Response: While the notion of dividing the paper into two distinct manuscripts is commendable, given the intricate interplay and simultaneous impact of Type 2 Diabetes Mellitus (T2DM)-related risk factors on both beta cells and infiltrating macrophages, ultimately contributing to beta cell loss and dysfunctionality, we have opted to retain all pertinent results within an unified manuscript.

  • The discussion requires further elaboration, especially when comparing it to other studies.

Response: Some parts were added to the discussion part to address provided comment.

  • Please enhance the quality of the figures; adding color to the graphs would be highly beneficial.

Response: The quality of Figure 12 has been improved. As the graphs were generated using Prism software, it is not feasible to introduce coloration.

  • Additionally, elucidate on the future prospects based on the study's conclusions.

Response: A segment has been incorporated into the discussion section to address these comments. Within this section, the limitations and potential future directions of the study have been delineated.

The introduction was modified.